# Changing Times: The Impact of Digitalization on the Behavior of Professionals and Their Perception towards Development

**DOI:** 10.3390/bs12050139

**Published:** 2022-05-10

**Authors:** Ariadna Badea, Nicolae Paun, Cristina Fleseriu, Dragos Paun

**Affiliations:** 1Doctoral School, Universitatea de Medicina si Farmacie “Iuliu Hatieganu”, 400012 Cluj-Napoca, Romania; badeaariadna@gmail.com; 2Institute of Global and Regional Studies, Universitatea Constantin Brancusi, 210185 Targu-Jiu, Romania; nicolae.paun@ubbcluj.ro; 3Department of Business, Universitatea Babes-Bolyai, 400174 Cluj-Napoca, Romania; cristina.fleseriu@ubbcluj.ro

**Keywords:** medical education, perception, behavior, costs

## Abstract

In 2020, the educational system was taken by surprise by the impact of the COVID-19 pandemic. Most of the educational institutions were delivering face-to-face classes and were forced to switch to online teaching in a very short period of time due to lockdown measures and the health and safety measures put in place by public authorities. In addition to universities, professional courses were also forced to be delivered online. Most of the time, these professional courses are important because they are directly linked to keeping the right to practice. The present paper focuses on the changing pattern in behavior of professionals and their acceptance of online courses. By applying over 1000 questionnaires in a timespan of more than one year, we have studied the impact of digitalization on the behavior and perception of professionals. We measured if the change towards online courses could be sustainable in the long run. The results of our study show that the behavior of professionals is different than those of students and that the online courses can be a long-term solution for education in professional environments.

## 1. Introduction

The COVID-19 pandemic has forced a lot of changes in the way we consider carrying out different activities. The novelty of this virus and its impact on our daily lives were unexpected. While certain parts of the world had similar experiences (MERS), most countries had to improvise and take measures to safeguard their citizens. Much of the literature in these years has focused on the COVID-19 pandemic, but the topics mostly refer to health issues and impacts on the economy. There are also numerous articles that focus on education, but they study students’ education and the impact that the pandemic had on them. Of course, formal education, represented by kindergartens, primary schools, secondary schools, colleges and universities, represents the backbone of the education system and part of them are mandatory in most countries. Looking at the number of impacted people, when it comes to education, these forms are the most visible. However, as these forms are the most visible, they are also the most formalized and coordinated by central authorities. Thus, changes can be implemented, and the reaction of the institutions can be sustained by the state via incentives or subsidies, as in most countries the educational institutions are financed by the state.

The present article does not focus on the formal education but looks at the education provided to professionals. While in most cases education is a milestone for qualification, and thus formal and centralized education stops after university, there are cases where the participation in educational programs is formally required throughout the career. There are initiatives for promoting lifelong learning which are supported by the European Union and states, and we also understand the importance of the professional development programs that are organized in every corporation or institution. However, the current article is focused on the courses that are required by law and that are mostly applied to liberal professions such as charted accountants, lawyers or doctors. The focus of our article is represented by the courses which are provided to doctors/physicians as these are the ones with the most impact on practical experience. While the courses designed for charted accountants or lawyers are centered around legislation, technical or soft skills (that are used in the profession and there is no need for demonstrations or showcase), the courses that are designed for doctors most of the time include demos and hands-on sessions. Thus, the authors of this study considered that we should focus our attention on the organization of these courses.

In Romania, the professional courses for liberal professions are highly regulated. In the case of accountants, auditors and lawyers, the courses are offered by the professional bodies and there are no third parties involved. Thus, we could consider that the quality of education cannot be contested, nor could there be an alternative in such a highly regulated environment. In the case of doctors, the professional bodies decentralized the right to offer professional courses and give special accreditations to noncommercial entities to organize such events. Thus, in this case, we see a liberalization and a competition for the organization of such courses. In the case of doctors, a decision by the Body of Doctors in Romania no. 12, published in the Official Gazette on the 6th of August 2018, CMR Nr. 12, requires each doctor to have 200 credit hours in a period of 5 years for the respective doctor to keep their license to practice. The activities for which these credit hours are given are represented by both offline and online courses, with 1 h of content representing 1 credit hour.

In Romania, in 2021, there were 159 bodies that were accredited by the Body of Doctors in Romania.

The literature analyzing the impact of COVID-19 on education is focused mainly on students and universities. Most of the articles present regional/local reports or studies conducted in a limited number of universities [1,2,3,4,5]. The focus on students in universities shows that there are categories of students that adopted online teaching and that would want to have this way in the future as well. However, this was the response of the students that were engaged in the program [3]. The same article also concludes that the preference towards online and offline studies is dependent on the time traveled. Another study from [4] looked at the interaction between students and teachers in higher education (HE) institutions in Hong Kong and revealed that students were satisfied with the online interaction, unlike the professors, who were not. The result of the study contradicts those of reference [6], reference [7] and reference [8], who concluded that students are unhappy about the lack of interaction in online classes. This lack of interaction and sometimes dissatisfaction could also be explained by the knowledge gap between students and professors. Studies have shown that educators had limited experience in collaborative tools or ICT [9,10]. The reference [5] show in a comparative study (professor and students) focusing on hospitality and tourism studies that there are differences between the studied groups. Professors do not believe that online education can be effective, as they are transformed into only content deliverers. We believe that this is not an issue, as in the case of the delivery of the professional courses, most of the lecturers are professionals, meaning that their time is limited and their focus is content delivery.

Of course, the feedback given by students/participants during the evaluation of the program is fundamental. The reference [11] present the critical importance and the biases of the evaluations in regard to teacher/professor. Of course, these evaluations could also include the impact of accessory channels, presence, timing, location and fatigue, but as the focus of this study were online courses, where participants could not be tracked, this was not the case. However, our study focused on the timing in terms of day of delivery and duration of the courses.

The impact and acceptance of online courses cannot be generalized to all fields of study. A study conducted in India by [1] shows that 70% of the students would opt for online classes, but when it came down to students from more technical fields such as agriculture, this was not the case. The study also reveals the fact that online studies allow students to learn at their own pace, from where they desire and in general allows for greater flexibility [12], in their research, looked at the perception of medical students towards online teaching, and their conclusion is different than that of the study conducted by [1]. In total, 57.1% of the respondents in the study found online teaching to be moderately useful and 49.8% considered that studying from home is not to be desired. When looking at the perspective of dentistry students, ref. [7] concluded in a study carried out in Indonesia that higher level students are dissatisfied with distance learning classes and that they prefer class learning for discussions. The challenges of distance learning were related to internet connection and its costs and difficulty to focus when learning online for a longer time. Ref. [13] conducted a study on dentistry students from India and concluded that 77% of the respondents strongly disagree with continuing in e-learning.

In Romania, the reference [14] researched the impact of the pandemic and online teaching on the training programs of medical residency. Their results highlight that 80% of the respondents reported an increase in the level of stress and anxiety, and 88.3% of them were dissatisfied with online courses. Among the reasons of dissatisfaction, the authors noticed the reduction in procedures during the COVID-19 pandemic and the reduction in access to patients, which is fundamental in the formation of doctors.

The reference [15] researched the impact of online and blended courses for healthcare professionals. The results of their study show that half of the participants were extremely satisfied with the content. The lack of studies focusing on healthcare professionals and experienced professionals in general led to the conceptualization of the current study. We, the authors, consider that there are many differences between the perception of students in university or early-stage professionals (residency students) and professionals. Among the factors that we must consider is that students are located in the vicinity of the universities during their studies, so there is proximity concerning their home and the training facilities. The students are in the stage of their career when the entire focus is on collecting information and learning. The availability of time is much higher in the case of students or early-stage professionals because most of the time revolves around their studies. The reference [16] measured the attitude of students concerning occupational medicine and their results show that after taking the course students actually realize the importance of all of the specialties. All of the considerations that we have included for students and early-stage professionals are not valid for experienced professionals. This category of people has already finalized its practical training and could not necessary be located around education institutions. The time for education activities is also limited as they must balance the time between jobs, family and other activities. Another factor that was reported by the literature as a reason for dissatisfaction is that the difference between the ICT competences between students and educators should not be valid in this case, as experienced professionals would be at the same level as educators. Specifically for Romania, the access to high-speed internet and the cost of the online infrastructure should not play a role in satisfaction, as Romania has one of the highest internet speeds and lowest prices, with a broadband connection of up to 1 GB costing around 20 EUR/month.

## 2. Materials and Methods

### 2.1. Study Design and Sample Description

Our study followed the perception of doctors participating in online and hybrid medical events during 2021 and 2022. An online questionnaire (Appendix A) was delivered to 1592 participants in 13 webinars and conferences. The questionnaire was created using Google Forms and was sent in the same email with the login information for the online and hybrid events. There were 1065 responses registered from doctors all over Romania.

The events (Table 1) that we have followed took place on Friday (in the case of the webinars), and over several days in the case of the hybrid conferences. The questionnaire can be found in Annex 1 of the present paper. The events that we followed had as lecturers (educators) experienced professors and professionals from medical institutions from Cluj-Napoca, Bucharest and Targu Mures. Most of the events are focused on ultrasound. The courses are focused on professionals, which we identified as a person engaged or qualified in a profession (in the case of this study, medical profession). The scope was also to measure the satisfaction and to estimate their future behavior in regard to these online courses. More specifically: Could such courses be viable after the pandemic? Is there a difference between professionals and the behavior of students, which many articles have focused on? The content that we followed in our study is related to professional development, but it is not a mandatory curriculum, thus, participants could have opted not to follow them.

### 2.2. Variables

The questionnaire was delivered at the end of the event and focused on the following ideas: if the location of the event was proper (online rather than offline); the satisfaction level of the participants; the impact of the event on their professional formation; the scientific quality of the presented content; the qualitative feedback of the event; the desired date in the week of future events; the type of event that the participants want to attend and the rationale of their participation. These variables were measured on a 10-point Likert scale where respondents rated their level of agreement with the statements in the questionnaire. No socio-demographic information was collected and the respondents were anonymous.

### 2.3. Data and Hypotheses

The data was first analyzed from a descriptive perspective. The median and the range were chosen as indicators because the evaluation scale is ordinal in nature. The Kolmogorov–Smirnov and Shapiro–Wilk tests were used to determine if the data are normally distributed. The results show that none of the tested variables have a normal distribution. When testing the hypothesis, the Kruskal–Wallis H test [17] was used as it is non-parametric. It also has the advantage of being more robust to outliers and it does not make assumptions about the distribution of data.

The hypotheses that were tested focused on if perceived bias influences the rating of the quality of the event:

**Hypothesis** **1** **(H1).**
*There is a significant difference in evaluation scores for the presented subjects between groups with different bias perception.*


**Hypothesis** **2** **(H2).**
*There is a significant difference in the perception of how much respondents learned between groups with different bias perception.*


**Hypothesis** **3** **(H3).**
*There is a significant difference in how much respondents would change their practice between groups with different bias perception.*


**Hypothesis** **4** **(H4).**
*There is a significant difference in evaluation scores for the organization of the event between groups with different bias perception.*


**Hypothesis** **5** **(H5).**
*There is a significant difference in evaluation scores for the perceived quality of the content between groups with different bias perception.*


**Hypothesis** **6** **(H6).**
*There is a significant difference in evaluation scores for the lecturers between groups with different bias perception.*


## 3. Results

Of the 1065 responses, 1018 were validated. Some missing data was also found for some of the variables, resulting in different samples depending on the analysis. The descriptive statistics for the sample are presented in Table 2.

As seen from Table 2, people considered that they learned something new (Item 1) with a median value of 10 for each event, even though for events 5 through 9 the values are more spread out, as indicated by the larger range and standard deviation and 11 presenting an outlier value. When asked if what they learned will change anything in their practice (Item 2), again the median response was 10, except for event 3 and 9 where the median was 9. The spread of evaluations was smaller for events 1, 4, 7 and 12, while event 9 again had the highest range because of an outlier value. These results are different than the results of [12], which questioned the satisfaction of students and found that 69% of the respondents were moderate or had low satisfaction. People agreed that the event was well organized (Item 3), all medians being 10 and presenting a small spread. These results are not supported by other studies [13] where the results show that 87% of students aged 22 were dissatisfied with the delivery of the courses in an online environment. The same is true for the quality of the scientific content (Item 4). The quality of the overall event (Item 5) was also highly evaluated by the respondents (median of 10) and people shared this opinion for each event (maximum range of 2), except for event 9 where the outlier value is again present. This information was not available for event 12. Because each event had different lecturers, a median was determined for all lecturers per each event. The results show that all lecturers were highly appreciated by the respondents. [6], in their research, found out that the universities were not prepared for the online teaching environment, and this could be one of the reasons for the quality of the content. Unlike in the case of universities, companies have the flexibility of investing in platforms so that the quality is acceptable for students.

The most preferred period for such an event is the end of the week (Figure 1). Half of the choices were for Friday and about 35% were for Saturday. Some respondents expressed preferences for multiple days.

Most respondents would rather take part in 4 to 7 h-long webinars (23%), but there is a close preference for the other types of events (Figure 2). A close second would be case studies and hybrid workshops where people would view online but watch how it is performed. These results are again in contradiction to the ones of [12], who questioned medical students, and their result was in favor of physical classes (67%). The results of [13] are also contrary to ours because in their case 73.8% of the students preferred offline classes. In our case, the respondents are in favor of webinars and case studies which are online and not hands-on, which would be physical.

The main reasons (52%) for taking part in such an event were the case studies that were presented and the commented images (Figure 3). The second most important aspect was the theoretical concepts (28%), while the collection of EMC points and the lecturers were not considered as important by most. The efficiency of these development programs (as seen from Figure 2 but also Table 2) are again in contradiction with the results of [14], who focused on residents and their opinion on online classes. In their studies, 88% of the respondents considered that online courses are not sufficient for their professional training. Again, we are considering here the fact that these differences are coming from students (even if they are in their last years) because a study by [15] also shows similar results to ours. In this study on healthcare professionals, half of the respondents were extremely satisfied with the online courses.

Most respondents (42%) found out about the event they attended from the platform’s newsletter (Figure 4). The second source of information was Facebook (22%) and the third was colleagues and friends (10%). The other sources were specific to the respondents’ fields.

Respondents were also asked if they felt that the event was biased. Most (75.6%) considered that the event was not biased, while 13.3% were neutral and 11.1% considered it biased.

When asked if they would pay a monthly subscription that would offer them access to all content, most preferred not to answer (55.9%), while 27.7% said “yes”. Of the 282 respondents that would be willing to pay, 53.9% would pay under EUR 50, 37.2% between EUR 50 and 60, while only 8.9% would pay more than this amount.

To test the first hypothesis (H1), a Kruskal–Wallis test was conducted to determine if there were differences in evaluation scores for the subjects between groups with different bias perception that perceived that the event was “biased”, “neutral” or “not biased”. Distributions of evaluation scores were not similar for all groups, as assessed by visual inspection of a boxplot. There was no statistical difference in evaluation scores between groups; H(2) = 2.648, N = 928 and *p* = 0.266.

The same test was conducted to test the second hypothesis (H2). Distributions of evaluation scores were not similar for all groups, as assessed by visual inspection of a boxplot. Median scores were statistically different between groups with different bias perception; H(2) = 7.241 and *p* = 0.027. Subsequently, pairwise comparisons were performed using [18] procedure with a Bonferroni correction for multiple comparisons. Adjusted *p*-values are presented. This post hoc analysis revealed statistically significant differences in evaluation scores between respondents that considered that the event was not biased (mean rank = 516.14) and respondents that remained neutral (mean rank = 462.38) (*p* = 0.026). Respondents that considered the event as not being biased had significantly higher scores than the ones that remained neutral.

The third hypothesis (H3) tested if there is a significant difference in how much respondents would change their practice between groups with different bias perception. The distribution of scores were again not similar for all groups, as assessed by visual inspection of a boxplot. Median scores were statistically different between groups with different bias perception; H(2) = 14.463 and *p* = 0.001. After applying a pairwise comparison using [18] procedure with a Bonferroni correction for multiple comparisons, statistically significant differences in evaluation scores were found between groups. Respondents that considered that the event was not biased (mean rank = 524.87) had significantly higher scores than respondents that remained neutral (mean rank = 434.09) (*p* = 0.001).

Hypothesis H4 tested if there is a significant difference in evaluation scores for the organization of the event between groups with different bias perception. Distributions of evaluation scores were not similar for all groups, as assessed by visual inspection of a boxplot. Median scores were statistically different between groups with different bias perception; H(2) = 10.177 and *p* = 0.006. Subsequently, pairwise comparisons were performed using Dunn’s (1964) procedure with a Bonferroni correction for multiple comparisons. The post hoc analysis revealed statistically significant differences in evaluation scores between respondents that considered that the event was not biased (mean rank = 519.17) and respondents that considered it biased (mean rank = 476.75) (*p* = 0.026). Respondents that considered the event as not being biased considered the event better organized than the ones that considered it biased.

Hypothesis H5 tested if there is a significant difference in evaluation scores for the perceived quality of the content between groups with different bias perception. The distribution of scores were again not similar for all groups, as assessed by visual inspection of a boxplot. Median scores were statistically different between groups with different bias perception; H(2) = 23.641 and *p* < 0.001. After applying a pairwise comparisons using Dunn’s [18] procedure with a Bonferroni correction for multiple comparisons, statistically significant differences in evaluation scores were found between groups. Respondents that considered that the event was not biased (mean rank = 518.83) (*p* < 0.001) and respondents that considered the event as being biased (mean rank = 519.96) had significantly higher scores than respondents that remained neutral (mean rank = 447.36) (*p* = 0.001).

The final hypothesis (H6) tested if there is a significant difference in evaluation scores for the lecturers between groups with different bias perception. The distribution of scores were assessed by visual inspection of a boxplot and are not similar for all groups. Median scores were statistically different between groups with different bias perception; H(2) = 30.509 and *p* < 0.001. After applying the same post hoc test, statistically significant differences in evaluation scores were found between groups. Respondents that considered that the event was not biased (mean rank = 520.80) (*p* < 0.001) and respondents that considered the event as being biased (mean rank = 516.51) had significantly higher scores than respondents that remained neutral (mean rank = 439.20) (*p* < 0.001).

## 4. Discussion

When assessing the results of our study, we see that they are contrary to the studies conducted by other researchers. This is due to the fact that the assumptions that were at the basis of students’ preference—reduction in costs and availability of time—are not validated in the case of professionals. As we see from the answers in the preferences of the organization of the events (Figure 1), professionals prefer days when they are off—meaning they are looking to develop themselves but outside the regular working hours. Moreover, when looking at the preferred type of event we notice that most of the respondents choose online events—such as webinars and case studies. The same results are shared by studies focusing on professionals [15,16].

Respondents’ rating of the material covered was independent of their belief about the bias of the course. This was not the case when looking at the perception of how much respondents learned. Respondents that considered the events as not being biased considered that they have learned much more than respondents that chose to remain neutral. This might indicate an unwillingness to express a negative opinion about the course. This phenomenon was also observed when asked about the quality of the content and of the lecturers, where people with a neutral opinion on bias rated both materials and lecturers significantly lower than people that expressed their (positive or negative) opinion on bias. Not surprisingly, respondents that considered the events as not being biased rated the overall quality of the event significantly higher than those that perceived the events as biased.

The results of our study are contradictory to studies that had a focus on university education. Refs. [3,6,19] concluded that online education is not so well-perceived and could lead to fatigue. The results of [9] are also opposite compared with our current study which highlights the benefits of online courses.

By focusing on the differences that other studies have pointed out, we can consider the quality of the materials [10,13]; the level of engagement [8,20] and the delivery method [7].

When assessing the overall quality of the programs, most of the literature that we analyzed presents different results than our study [1,3,7,8,12] with dissatisfaction ranging from 88% in the case of [14] to 5% in the case of [8]. However, there are studies that focus on professionals in which the results are similar to ours [15]. In the case of the research by [15], half of the respondents considered these courses to be “extremely efficient”. The fundamental reason why all of the studies have opposite results is their focus on university education. Professional development and professionals are more focused on their career and see the benefits of distance learning. This phenomenon could also be seen from the age of students that are following part-time, distance or online learning programs, even in universities. All these reasons led us to the conclusion that these types of online courses have a future and that the pattern of behavior for professionals is not the same as that of students.

The study presents some limitations, as also mentioned in the text. One of the limitations is the fact that it followed only one online education provider. Thus, we could say that the perception and impact could be due to the quality of the organizer. A different level of quality could not have the same impact on consumers, and thus further research is needed. However, the results of the present study show clearly that there is a change in behavior and that digitalization could improve this process. Another aspect that was not evaluated during this study is how the participants perceived the lack of physical interaction with the trainers and other participants. This is an important aspect of online and hybrid education that should be evaluated in future studies.

Another limitation is that the authors did not measure the impact of accessory channels such as location, fatigue and lack of interaction between participants, although some respondents replied in their open-answer question that fatigue starts to settle in after several hours. In this respect, further analysis must be conducted so that the authors could establish the optimum duration of online courses.

## 5. Conclusions

In the current article, the authors wanted to test the perception of professionals that participated in online courses during 2021 and 2022. The results of the analysis show that the introduction of online courses has been beneficial and that there is stability towards online education. These results are in contradiction with the results of previous studies conducted on university students but are in line with other studies that focus on professionals. Of course, even though the research is based on a large sample of results, 1592 people were asked to review these courses and 1065 answers were reported, there are still limitations to the research. The research followed just one provider of online courses, and even though the responses were from participations from all regions of Romania, there still is the influence of the quality of this single provider. Future research should be conducted by including other online providers to see if there is a preference for one provider or if the perception is related to the method of delivery of the content.

## Figures and Tables

**Figure 1 behavsci-12-00139-f001:**
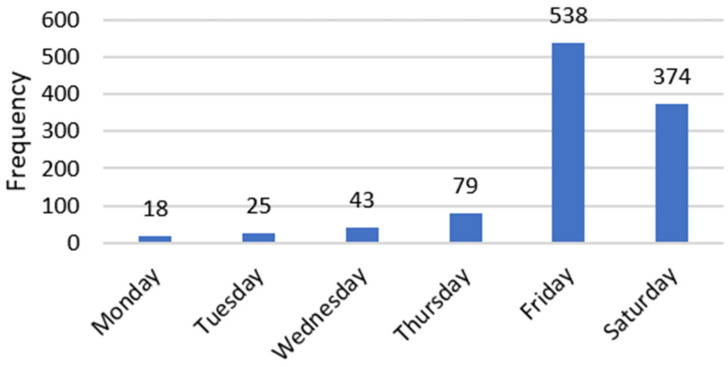
Preferred day of the week for taking part in an online professional course.

**Figure 2 behavsci-12-00139-f002:**
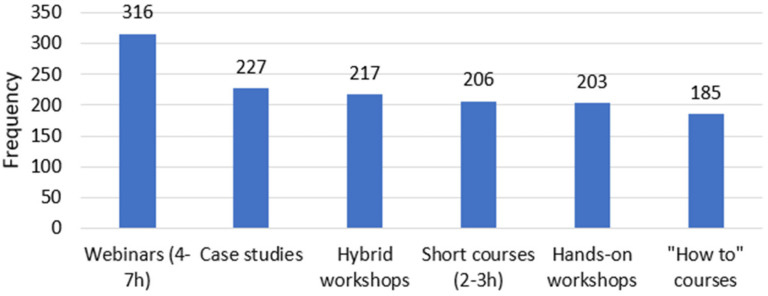
Preferred type of event by respondents.

**Figure 3 behavsci-12-00139-f003:**
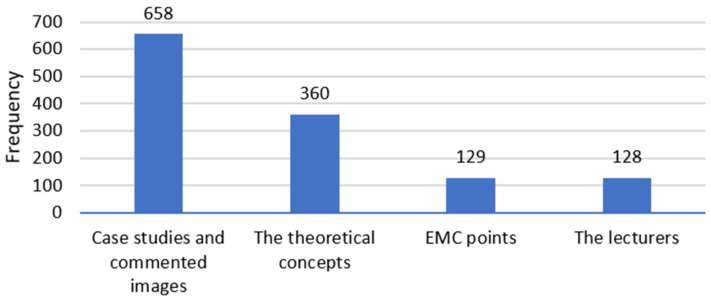
The main reasons for attending an online professional course.

**Figure 4 behavsci-12-00139-f004:**
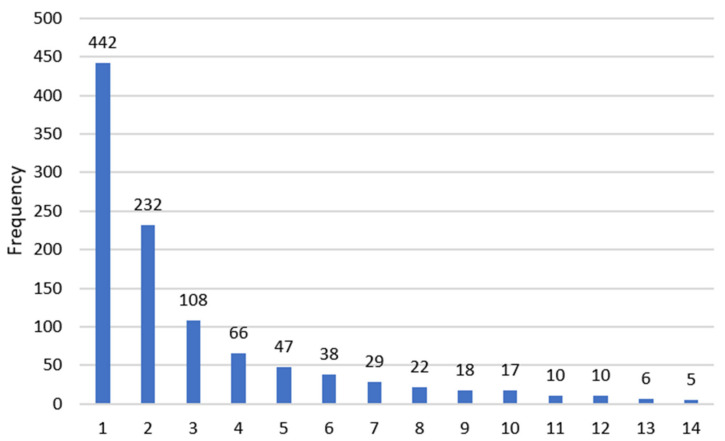
The main sources of information regarding such an event. 1—The Eduson Newsletter; 2—Facebook; 3—Friends/Colleagues; 4—SRUMB; 5—TopMed; 6—SOGR; 7—Eduson.ro; 8—Other; 9—Colegiul Medicilor; 10—ARSUS; 11—The institution where I work; 12—Asociatia Medicilor de Familie; 13—Colegiul Fizioterapeuțillor din România; 14—Asociatia pentru Ecografie Mures.

**Table 1 behavsci-12-00139-t001:** Events tracked during the study.

Name	Date	Participants
Eduson Conference	11–12 December 2021	127
Fetal Ultrasound	4 March 2022	109
3rd CEUS Webinar	25 February 2022	34
Postural Issues	18 February 2022	104
Ultrasound for Family Doctors 3rd Webinar	11 February 2022	107
2nd CEUS Webinar	21 January 2022	56
Ultrasonography of the veins of the lower limbs	28 January 2022	108
Ultrasound for Family Doctors 2nd Webinar	14 January 2022	114
Ultrasound for Pediatricians	7 January 2022	206
TopMed Conference	15–17 November 2021	240
Ultrasound for Gynecology	14 December 2021	77
1st CEUS Webinar	10 December 2021	40
Muskuloscheletal Ultrasound	25 November 2021	270

Authors own calculation based on data provided by the organizers of the events.

**Table 2 behavsci-12-00139-t002:** Descriptive statistics for the sample.

Item	Statistic	Event											
		1	2	3	4	5	6	7	8	9	10	11	12
1. Did I learn anything from this event?	N	43	28	32	20	73	26	46	42	127	34	120	146
	Mean	9.86	9.46	9.72	9.65	9.6	9.65	9.76	9.67	9.56	9.79	9.59	9.75
	Median	10	10	10	10	10	10	10	10	10	10	10	10
	St. dev.	0.47	0.92	0.52	0.59	0.80	0.89	0.64	0.72	0.72	0.48	1.08	0.56
	Min	8	7	8	8	6	6	7	7	7	8	1	7
	Max	10	10	10	10	10	10	10	10	10	10	10	10
	Range	2	3	2	2	4	4	3	3	3	2	9	3
2. Will this event change my practice?	N	43	28	32	20	73	26	46	42	127	34	120	146
	Mean	9.63	9.11	8.94	9.15	9.21	9.04	9.57	9.57	8.71	9.68	9.27	9.40
	Median	10	10	9	10	10	10	10	10	9	10	10	10
	St. dev.	0.76	1.26	1.44	1.09	1.26	1.54	0.86	1.17	1.63	0.84	1.35	0.91
	Min	7	5	4	7	5	5	6	3	3	6	1	6
	Max	10	10	10	10	10	10	10	10	10	10	10	10
	Range	3	5	6	3	5	5	4	7	7	4	9	4
3. Was the event well organized?	N	43	28	32	20	73	26	46	42	127	34	120	146
	Mean	9.93	9.64	9.94	9.9	9.85	9.96	9.87	9.76	9.86	9.79	9.84	9.86
	Median	10	10	10	10	10	10	10	10	10	10	10	10
	St. dev.	0.26	0.68	0.25	0.31	0.52	0.20	0.40	0.58	0.41	0.73	0.50	0.46
	Min	9	8	9	9	7	9	8	8	8	6	7	7
	Max	10	10	10	10	10	10	10	10	10	10	10	10
	Range	1	2	1	1	3	1	2	2	2	4	3	3
4. How would you rate the scientific content?	N	43	28	32	20	73	26	46	42	127	34	120	146
	Mean	10	9.61	9.84	10.00	9.92	9.81	9.93	9.88	9.80	9.91	9.88	9.89
	Median	10	10	10	10	10	10	10	10	10	10	10	10
	St. dev.	0.00	1.03	0.45	0.00	0.32	0.80	0.25	0.33	0.49	0.38	0.46	0.36
	Min	10	5	8	10	8	6	9	9	8	8	7	8
	Max	10	10	10	10	10	10	10	10	10	10	10	10
	Range	0	5	2	0	2	4	1	1	2	2	3	2
5. Was the venue appropriate?	N	43	28	32	20	73	26	46	42	127	34	120	146
	Mean	10.00	9.61	9.84	9.85	9.95	9.81	10.00	9.76	9.78	9.68	9.81	N/A
	Median	10	10	10	10	10	10	10	10	10	10	10	N/A
	St. dev.	0.00	0.63	0.45	0.37	0.23	0.49	0.00	0.58	0.52	1.57	0.52	N/A
	Min	10	8	8	9	9	8	10	8	8	1	8	N/A
	Max	10	10	10	10	10	10	10	10	10	10	10	N/A
	Range	0	2	2	1	1	2	0	2	2	9	2	N/A
6. Lecturer evaluation	N	43	28	32	20	73	26	46	42	127	34	120	146
	Mean	9.95	9.79	9.91	9.90	9.92	9.85	9.94	9.95	9.75	9.90	9.90	9.95
	Median	10	10	10	10	10	10	10	10	10	10	10	10
	St. dev.	0.21	0.50	0.30	0.31	0.25	0.61	0.25	0.22	0.62	0.52	0.36	0.24
	Min	9	8	9	9	8.5	7	9	9	6	7	8	8
	Max	10	10	10	10	10	10	10	10	10	10	10	10
	Range	1	2	1	1	1.5	3	1	1	4	3	2	2

1—Ultrasound for Family Doctors 2nd Webinar; 2—CEUS; 3—CEUS 2; 4—CEUS 3; 5—Fetal Ultrasound; 6—Postural Issues; 7—Ultrasonografia etajului tubului digestiv și a spațiului retroperitoneal; 8—Ultrasonography of the Veins of the Lower Limbs; 9—Eduson Conference; 10—Ultrasound for Gynecology; 11—Ultrasound for Pediatricians; 12—Muskuloscheletal Ultrasound.

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
