# Peer review of "Changing Times: The Impact of Digitalization on the Behavior of Professionals and Their Perception towards Development"

_behavsci, 2022, doi:10.3390/bs12050139_

Round 1

Reviewer 1 Report

Dear Editor,
I really appreciate the opportunity to review the manuscript behavsci-1697979 entitled:
"Changing times: The impact of digitalization in the behavior of professionals and their perception towards development"

I commend the authors for describing this critical and timely issue. The paper is interesting and well-written; however, I would like to highlight some issues that merit revision:

The article comes across as particularly interesting and the study certainly appears to be well conducted. I did not find references in the evaluations conducted to how the lack of physical interaction with event participants is perceived. It is well known that human communication does not pass only through verbal and semantic channels but to a large extent also through accessory channels, presence, timing, location, fatigue etc. I ask the users if this has been evaluated and if so, to briefly describe it in the text. If it has not been evaluated, please add the lack of these aspects in the limitations, since the spread of online updates is likely to limit the training in its component of inter-individual interaction.

Author Response

First of all we want to thank the reviewer for his / her kind words! We are really happy that our work is considered interesting and we hope it will be read by many researchers once published. We also want to thank the reviewer for pointing out how we could improve our article. We did not research the impact of accessory channels, we only studied the timing of such courses (during which day would the respondents prefer to have the courses). As such we have taken the recommendation of the reviewer and we have inserted a paragraph specific to accessory channels in the limitations of the study. (line 559). 

Reviewer 2 Report

The topic covered in the article is interesting and very topical. However, there are some aspects that, in my opinion, should be clarified before being published:
1. It would be advisable for the authors to make a thorough revision of their manuscript to correct typographical or expression errors. For example, in line 23 there is an extra semicolon, in line 71 the paragraph is abruptly cut off, in line 153 it says Kruskar instead of Kruskal, and in line 204 it says testes instead of tested.
2. it is not clear to me what specifically the authors mean by "behavior of professionals" and "development", which should be influenced by digitization. This should be clarified by defining specific variables in the Materials and Methods section. 
3. The Material and Methods section needs to be restructured, in my opinion. Sub-sections should be introduced to explain the characteristics of the participants, the variables measured, the hypotheses tested (in the current version of the manuscript, these are indicated in the Results section) and the instruments used, as well as the procedure followed.
4. Table 2 is unnumbered and untitled. In addition, I advise including this table as an appendix to facilitate reading.
5. It is not clear, as far as I have been able to understand, why the authors have chosen a nonparametric test for the hypothesis test involving comparison of means, instead of a parametric test. Perhaps the authors have applied a normality test and concluded that the distributions are not normal? The authors suggest something about this in lines 151 to 155, but it would be useful to make it more explicit in the Materials and Methods section.
6. The authors should improve the aesthetic quality of the graphs (Figures 1 to 4). In particular, I recommend putting the text in black, including vertical and horizontal axes and removing the outer frame. Figure 4 should also be reworked to make it more legible.
7. The Introduction should be strengthened by referencing previous literature that addresses questions about the impact of the digitization of learning environments. This will allow the authors to strengthen the Discussion. I propose, as an example, some current references in this regard:
https://doi.org/10.3390/ijerph19063732
https://doi.org/10.1080/10963758.2021.1907196
https://doi.org/10.1097/MD.0000000000027684
8. The Discussion still has ample room for improvement. In particular, the authors should summarize the results obtained and relate them to the previous literature.

Author Response

First of all we want to thank the reviewer for taking the time to read our manuscript and to review it. We believe that the recommendation have made this article much better and we hope we have answered all of them. 

As such here are the answers to the review and of course these are also visible in the text which has been uploaded: 

  1. It would be advisable for the authors to make a thorough revision of their manuscript to correct typographical or expression errors. For example, in line 23 there is an extra semicolon, in line 71 the paragraph is abruptly cut off, in line 153 it says Kruskar instead of Kruskal, and in line 204 it says testes instead of tested.

The manuscript was revised by a third party and all found error were corrected.

  1. it is not clear to me what specifically the authors mean by "behavior of professionals" and "development", which should be influenced by digitization. This should be clarified by defining specific variables in the Materials and Methods section. 

We have explained a bit more clearer these issues in the text. What we are considering to be professionals and why it was important to review the differences between the behavior and acceptance of such courses for this category compared to university students and professions on which the majority of the literature is focused. 

  1. The Material and Methods section needs to be restructured, in my opinion. Sub-sections should be introduced to explain the characteristics of the participants, the variables measured, the hypotheses tested (in the current version of the manuscript, these are indicated in the Results section) and the instruments used, as well as the procedure followed.

The Materials and Methods section was restructured. The hypotheses were moved to the Materials and Methods section.

  1. Table 2 is unnumbered and untitled. In addition, I advise including this table as an appendix to facilitate reading.

The table was numbered and the caption was added. If the reviewer and editor consider this can also be moved in the appendix. 

  1. It is not clear, as far as I have been able to understand, why the authors have chosen a nonparametric test for the hypothesis test involving comparison of means, instead of a parametric test. Perhaps the authors have applied a normality test and concluded that the distributions are not normal? The authors suggest something about this in lines 151 to 155, but it would be useful to make it more explicit in the Materials and Methods section.

Indeed, the data is non normally distributed and that is why non-parametric tests were used. This fact was more explicitly stated in the Materials and Methods section.

  1. The authors should improve the aesthetic quality of the graphs (Figures 1 to 4). In particular, I recommend putting the text in black, including vertical and horizontal axes and removing the outer frame. Figure 4 should also be reworked to make it more legible.

The figures were improved to make them more legible as per the reviewer’s recommendations.

  1. The Introduction should be strengthened by referencing previous literature that addresses questions about the impact of the digitization of learning environments. This will allow the authors to strengthen the Discussion. I propose, as an example, some current references in this regard:
    https://doi.org/10.3390/ijerph19063732
    https://doi.org/10.1080/10963758.2021.1907196
    https://doi.org/10.1097/MD.0000000000027684

We thank the reviewer for these suggestions. These articles have been taken into consideration and included together with others in the manuscript.

  1. The Discussion still has ample room for improvement. In particular, the authors should summarize the results obtained and relate them to the previous literature.

We thank the reviewer for tis suggestion. Indeed we have reviewed the discussion part and we hope that it can be considered suitable for acceptance. 

Again we want to thank the reviewer for the suggestions. Upon reading the manuscript after the changes we believed that the quality as improved considerably. We are open for other suggestions because every idea has improved the quality of our work and will make the manuscript more relevant for the literature. 

Round 2

Reviewer 2 Report

The authors have made important modifications to their manuscript that have undoubtedly increased the expository and structural quality of the article. However, there is still, in my opinion, a problem of scarcity of references that makes the discussion of the results weak. I recommend that the authors review this issue, following the comment I made in my previous report.

Author Response

Thank you for the suggestions. We have added different references to previous studies in both the discussion section and in the results section. In this way readers can compare the results of this current study with other similar studies. We hope that the current discussion is satisfactory but if we need to make further changes we would kindly ask the reviewer for more guidance in this respect. 

Round 3

Reviewer 2 Report

I thank the authors for their efforts to improve the quality of their manuscript. They have responded to the issues raised by me in a satisfactory manner.